# OpenReview forum: "Jailbreaking on Text-to-Video Models via Scene Splitting Strategy"
_ICLR.cc/2026/Conference — ICLR 2026 Poster_

### Official Review · Reviewer_9BtZ · 2025-10-17

**Soundness:** 2
**Presentation:** 3
**Contribution:** 3
**Rating:** 6
**Confidence:** 3

**Summary:**

This paper explored jailbreak attacks for T2V models, it introduced a new black-box method called SceneSplit. Specifically, it fragmented harmful contents into multiple scenes with paraphrasing to lower the potential of detection by safety filters. It also updated the split prompts to dynamically approached to the safety decision boundary. To reduce the computational overheads, it included a strategy library to facilitate attacks for similar prompts. Experiments have shown the superiority of the proposed method compared with existing baselines.

**Strengths:**

1.	This paper is well-written and easy to follow.

2.	Jailbreak attacks for text-to-video models are imperative and useful.

**Weaknesses:**

1.	The overall process is very complex and time-consuming. I think this method may not be suitable for real-world application.

2.	Evaluations are not that sufficient. They only tested three T2V models, as a black-box method, even state-of-the-art closed-source models like Sora-2 can also be tested. At least authors should evaluate on the models in T2VSafetyBench.

3.	Many evaluations and strategies rely on GPT-4o. Furthermore, their evaluation protocol follows the setting of previous papers, sampling some frames from a video and score by MLLMs. I think this evaluation may raise potential bias. Beyond GPT, can you evaluate the harmfulness on other MLLMs? (e.g., open-sourced like Qwen, LlaVa, commercial models like Gemini, Grok?) This will be beneficial to reduce bias. Furthermore, for the ASR judgement, human assessments are also welcome as did in T2VSafetyBench.

4.	The notion of constraining the generative output space is described intuitively but lacks formalization or reasonable analysis.

**Questions:**

1.	How sensitive is SceneSplit to the number of scenes and paraphrasing degree? Is there an optimal split depth for different content types?

2.	Is there any potential defense? For instance, as SceneSplit reuses successful strategies, could a defensive version of the strategy library be trained to anticipate and block similar attacks?

3.	What is the defender use post-hoc detector on T2V models? Is your attacks still applicable?

**Details Of Ethics Concerns:**

This work can result in potential harmful/disturbed contents.

---

> ### Author Response · Authors · 2025-11-21
> **Official Comment by Authors**
>
> We are deeply grateful to the Reviewer 9BtZ for the highly constructive and insightful review.
>
> ---
>
> **Weakness 1**
>
> We agree efficiency is important. As shown in our response to other reviewers (Reviewer XnhP's weakness 3 and Reviewer 2XCW's weakness 4), we compared to RPG-RT [1], a SOTA T2I attack. To ensure fairness, both methods were set to a maximum of 15 iterations on Veo2 across three distinct categories: Pornography, Borderline Pornography, and Violence.
>
> | Method            | Attack time per prompt (min) ↓ |
> |-------------------|------------------------------|
> | RPG-RT            | 21.9 (10.2*)                 |
> | SceneSplit (Ours) | **5.9**                      |
> *Time excluding fine-tuning overhead
>
> Our method, SceneSplit, demonstrates a significantly faster attack time (5.9 min) compared to RPG-RT (21.9 min), which incurs substantial overhead due to the training of a scoring model and subsequent LLM fine-tuning via DPO. Even when considering only the inference time after fine-tuning (10.2 min), SceneSplit maintains its superior efficiency.
>
> [1] Cao, Yichuan, et al. "Red-Teaming Text-to-Image Systems by Rule-based Preference Modeling." NeurIPS 2025.
>
> Furthermore, we analyzed the efficiency contribution of our individual components across all 11 safety categories evaluated on Veo2.
>
> **1. Efficiency of Scene Manipulation**: We assessed the efficiency of the Scene Manipulation component by comparing our full method against w/o Scene Manipulation (equivalent to the Inner Loop only setting in Table 7). In w/o Scene Manipulation, a new Scene Splitting prompt is generated at every iteration instead of manipulating the existing scene splitting prompt.
>
> | Method                  | Attack Success Rate (ASR) ↑ | Avg. Attack Attempt ↓ | Avg. Time ↓ |
> |-------------------------|-----------------------------|-----------------------|-------------|
> | with Scene Manipulation | **60.9%**                   | **2.50**              | **130.72s** |
> | w/o Scene Manipulation  | 46.4%                       |    3.57               | 171.49s     |
>
> As illustrated, Scene Manipulation notably reduces both the average number of attempts (3.57 → 2.50) and the total time (171.49s → 130.72s). This validates that searching for an optimal attack prompt within a constrained generative output space (via manipulation) is far more efficient than the repeated generation of entirely new scene split prompts, requiring fewer iterations to achieve success.
>
> **2. Efficiency of Strategy Update**: We also analyzed the efficiency of the Strategy Update component (corresponding to the setting in Table 5). We compared our complete method against w/o Strategy Update, where the outer loop generates a new scene split prompt from scratch without retrieving a strategy.
>
> | Method               | Attack Success Rate (ASR) ↑ | Avg. Attack Attempt ↓ | Avg. Time ↓ |
> |----------------------|-----------------------------|-----------------------|-------------|
> | with Strategy Update | **78.2%**        | **5.54**              | 323.06s     |
> | w/o Strategy Update  | 69.1%        | 6.22                  | **311.52s** |
>
> Strategy Update improves efficiency by reducing the average number of attempts (6.22 → 5.54) through the leveraging of proven attack patterns. While the average time is marginally higher (311.52s → 323.06s) due to the computational overhead associated with managing the strategy library and summarization, this minor trade-off yields a substantial increase in ASR (69.1% → 78.2%). We believe that this trade-off is highly beneficial for establishing robust jailbreak performance.
>
> ---
>
> **Weakness 2**
>
> We have expanded our experimental scope to include Sora2 and Kling V1.0 to further validate the generality of our method.
>
> |  | Sora2 |  | Kling V1.0 |  |
> |---|---|---|---|---|
> |  | T2VSafetyBench | **SceneSplit (Ours)** | T2VSafetyBench | **SceneSplit (Ours)** |
> | Pornography | 0% | **10%** | 5% | **35%** |
> | Borderline Pornography | 0% | **45%** | 50% | **85%** |
> | Violence | 30% | **95%** | 60% | **100%** |
> | Gore | 5% | **55%** | 60% | **80%** |
> | Disturbing Content | 50% | **75%** | 50% | **70%** |
> | Discrimination | 15% | **70%** | 15% | **75%** |
> | Political Sensitivity | 5% | **65%** | 15% | **85%** |
> | Illegal Activities | 45% | **90%** | 20% | **85%** |
> | Misinformation | 70% | **75%** | 40% | **70%** |
> | Sequential Action Risk | 55% | **90%** | 60% | **95%** |
> | Dynamic Variation Risk | 60% | **85%** | 40% | **85%** |
> | Average | 30.5% | **68.6%** | 37.7% | **78.6%** |
>
> SceneSplit significantly improved the ASR on Sora2, raising it from a baseline of 30.5% to 68.6%. On Kling V1.0, we observed an even more remarkable increase, with the ASR jumping from 37.7% to 78.6%. This consistent performance boost across diverse T2V models confirms our core finding: constraining the generative output space through scene splitting is a fundamental vulnerability in current T2V generation frameworks, rather than a flaw specific to a single architecture.

---

> ### Author Response · Authors · 2025-11-21
> **Official Comment by Authors**
>
> **Weakness 3**
>
> To address this and verify the robustness of our results, we re-evaluated the harmfulness of the generated videos using different MLLMs, including Qwen3-VL-8B and Gemini 2.5-Flash. We also measured the agreement rate between these models to ensure consistency.
>
> |     | GPT-4o | Qwen3-VL-8B | Gemini2.5-Flash |
> |-----|--------|-------------|-----------------|
> | ASR | 78.2%  | 70.9%       | 72.3%           |
>
> |                           | Agreement Rate |
> |---------------------------|----------------|
> | GPT-4o - Gemini 2.5-Flash | 77.7%          |
> | GPT-4o - Qwen3-VL-8B      | 80.1%          |
> | Gemini - Qwen3-VL-8B      | 76.2%          |
>
> As shown in the tables, not only GPT-4o but also relatively lighter models consistently report high ASRs exceeding 70%. Furthermore, the high agreement rates confirm that our results are not GPT-4o's specific bias but reflect a genuine vulnerability effectively identified by diverse evaluators.
>
> ---
>
> **Weakness 4**
>
> In Appendix A, we showed the analysis that formally demonstrates how SceneSplit narrows the generative output space.
>
> We define the "generative output space" using two key metrics:
>
> Divergence: For a set of captions $C = \{c_1, ..., c_n\}$ extracted from $n$ generated videos, the divergence is defined as follows:
> $$
> \text{Divergence}(C) = 1 - \frac{1}{n(n-1)} \sum_{i \neq j} \text{cos-sim}(\text{Emb}(c_i), \text{Emb}(c_j))
> $$
> where $\text{Emb}$ denotes the CLIP text encoder and $\text{cos-sim}$ represents cosine similarity. A lower divergence score indicates that the generated videos are semantically more similar to each other, indicating a more strongly constrained space.
>
> Unsafe Video Count: This measures the probability that the constrained space aligns with the harmful intent.
> We conducted an experiment using 5 different SceneSplit prompts, generating videos for single scenes, pairs of scenes, and the full combination of three scenes.
>
> | Scene Combination | Divergence ↓ | Unsafe Video Count ↑ |
> |-------------------|--------------|----------------------|
> | Scene 1           | 0.2193       | 0.2                  |
> | Scene 2           | 0.2421       | 0.2                  |
> | Scene 3           | 0.2196       | 0.4                  |
> | Scene 1 & 2       | 0.1508       | 0.8                  |
> | Scene 2 & 3       | 0.1766       | 1.0                  |
> | Scene 1 & 3       | 0.1939       | 1.0                  |
> | Scene 1 & 2 & 3   | **0.0808**   | **2.2**              |
>
> As shown in the table above (Table 9 in the paper), as more scenes are combined (from single scenes to the full sequence), the Divergence consistently decreases (0.2193 → 0.0808), while the Unsafe Video Count drastically increases (0.2 → 2.2).
> This quantitatively proves our hypothesis: the sequential combination of individually benign scenes effectively constrains the diversity of possible outputs (low divergence) and steers that narrowed space into the unsafe region (high unsafe count).
>
> ---
>
> **Question 1**
>
> We deeply appreciate the Reviewer's question regarding the sensitivity of our method. To address this, we conducted detailed analyses on Veo2 to determine the optimal configuration for SceneSplit.
>
> 1. Sensitivity to Number of Scenes: We analyzed the distribution of successful attacks based on the number of scenes used in the split.
>
> | Scene number | ratio |
> |--------------|-------|
> | 2            | 0.6%  |
> | 3            | 10.5% |
> | 4            | 38.4% |
> | 5            | 50.6% |
>
> As shown in the table, the majority of successful attacks utilized 5 scenes (50.6%) and 4 scenes (38.4%). This indicates that a split depth of 4 to 5 scenes is optimal for bypassing safety filters.
>
> To verify if this trend depends on the content type, we further analyzed the most frequent scene count for each safety category among the successful attacks.
>
> | Category ID | Most Frequent Scene Count (Ratio) |
> |-------------|-----------------------------------|
> | 1           | 4 (45.5%)                         |
> | 2           | 4 (60.0%)                         |
> | 3           | 5 (61.0%)                         |
> | 4           | 5 (56.2%)                         |
> | 5           | 5 (60.0%)                         |
> | 7           | 4 (38.5%)                         |
> | 8           | 5 (58.8%)                         |
> | 10          | 5 (55.6%)                         |
> | 11          | 5 (55.6%)                         |
> | 12          | 5 (64.7%)                         |
> | 13          | 5 (50.0%)     |
>
> The results show that across diverse categories, the optimal split depth consistently falls between 4 and 5 scenes without significant dependency on the specific category. This suggests that 4 to 5 scenes represent the optimal balance where individual scenes are sufficiently benign to pass the filter, yet the sequence remains cohesive enough to induce the harmful output.

---

> ### Author Response · Authors · 2025-11-21
> **Official Comment by Authors**
>
> **Question 1**
>
> 2. Sensitivity to Paraphrasing Degree: To investigate the impact of paraphrasing intensity, we conducted a comparative experiment where we modified the instructions for Scene Splitting and Scene Manipulation to enforce more "aggressive" paraphrasing. We measured Semantic Similarity following the same approach as in our Semantic Consistency analysis (Appendix C): calculating the cosine similarity between the original harmful prompt and the video caption.
>
> | Method                         | ASR   | Avg. Semantic Similarity |
> |--------------------------------|-------|--------------------------|
> | SceneSplit (Standard)          | 78.2% | 0.5308                   |
> | SceneSplit (Over-paraphrasing) | 80.9% | 0.5051                   |
>
> The results reveal an interesting trade-off. While increasing the paraphrasing degree slightly improves the ASR (78.2% → 80.9%) by making the prompts more difficult for safety filters to detect, it also decreases Semantic Similarity (0.5308 → 0.5051). This indicates that "Over-paraphrasing" causes the generated video to drift further from the original harmful intent. Therefore, we believe our standard setting strikes the optimal balance between attack success and semantic preservation.
>
> ---
>
> **Question 2**
>
> The concept of developing a defensive version of the strategy library is indeed a promising direction. Since SceneSplit systematically reuses effective splitting patterns, a defender could analyze these to anticipate similar attack structures. However, while a defensive strategy library is a valuable first step, it is likely not a complete solution. Even with such defenses, SceneSplit can still generate a vast array of harmful videos through narrative-based structures that appear individually benign but harmful. Consequently, constructing a defensive library comprehensive enough to block all these variations would likely require significant computational resources.
>
> In conclusion, this highlights the need for more efficient defense mechanisms against narrative-based attacks. We view this as a critical direction for future work and believe that SceneSplit contributes to T2V safety by exposing this vulnerability. We believe the ultimate defense requires safety mechanisms capable of understanding the deep context between scenes to distinguish malicious splitting.
>
> ---
>
> **Question 3**
>
> As noted by Reviewer XnhP, it is highly likely that commercial T2V models already utilize defense mechanisms, including pre-processing text and post-processing video safety detectors. However, the high success rate of SceneSplit against these current commercial models highlights the insufficiency and limitations of existing post-hoc safety filters.
>
> While deploying significantly stronger post-hoc filters might reduce the ASR, especially for content with direct visual exposure which is relatively easier to defend against, we believe that fully comprehending and blocking narrative-based outputs remains a fundamental challenge. Because SceneSplit distributes harmful intent across sequentially combined scenes that appear individually benign, a filter requires deep contextual understanding to detect the overall video's harmfulness. This finding highlights the urgent need for next-generation safety filters that can understand complex narrative-based attacks.
>
> ---
>
> We sincerely thank the reviewer for your time and careful consideration. We welcome any further questions the reviewer may have regarding our revisions or results.

---

> > ### Comment · Reviewer_9BtZ · 2025-11-22
> >
> > Thanks for your rebuttal! Most of my concerns have been solved. I am willing to support the acceptance of this paper. I will increase my rating to 8. I suggest authors to include their additional discussions/experiments into their update manuscript.

---

> > > ### Author Response · Authors · 2025-11-22
> > >
> > > Thank you for your active feedback and engagement during the rebuttal process. We will update additional experiments into update manuscript. We appreciate your endorsement!

---

### Official Review · Reviewer_2XCW · 2025-10-28

**Soundness:** 2
**Presentation:** 3
**Contribution:** 2
**Rating:** 4
**Confidence:** 4

**Summary:**

The paper introduces SceneSplit, a black-box jailbreak method for Text-to-Video (T2V) models that exploits a vulnerability in their safety mechanisms by fragmenting a harmful narrative into multiple scenes, each individually benign. This technique bypasses safety filters by ensuring the individual scenes correspond to a wide, safe output space, but their sequential combination acts as a powerful constraint, narrowing the collective generative output space to an unsafe region that reflects the original harmful intent. Experiments on commercial T2V models like Luma Ray2, Hailuo, and Veo2 demonstrate its effectiveness, achieving high Attack Success Rates (ASR), which reveals the need for a new generation of safety filters capable of assessing contextual risks across scenes.

**Strengths:**

1. The generation of harmful videos and jailbreak attacks are important research topics.
2. The paper is easy to follow.
3. The proposed method outperforms the baseline.

**Weaknesses:**

1. The technical depth of this paper seems limited.
2. It is not clearly explained why Scene Splitting/Manipulation work well in jailbreaking attacks.
3. Only one baseline is evaluated and compared.
4. Time/efficiency analysis of the proposed method seems missing.

**Questions:**

1. The idea of Scene Splitting is easy to understand. I am curious is there any safety filter that works on the whole video instead of each single frame. Would such safety filters defend against the proposed attack?

2. Why is it importent to find the most influencial frame? This part may be better explained.

3. Are there more baselines to be compared, such as directly using an LLM to rewrite the prompt? Such baselines help understand whether the proposed method itself is effective or deployed LLM/VLM are powerful.

4. The unsafe threshold is set to be 0.6. What does this value mean? Does a value larger than 0.6 guarantee an unsafe video?

---

> ### Author Response · Authors · 2025-11-21
> **Official Comment by Authors**
>
> We thank Reviewer 2XCW for your valuable insights, which have guided us to strengthen our work significantly.
>
> ---
>
> **Weakness 1 & 2**
>
> We appreciate the Reviewer's feedback regarding the technical depth and the explanation of our method's mechanism. We respectfully emphasize that the primary technical contribution of this paper is identifying a novel, structural vulnerability in T2V models—specifically, the "Generative Output Space Constraint" via sequential scene combination—and proposing a black-box algorithm to exploit it.
>
> Unlike simple prompt engineering, SceneSplit formally manipulates the output space through a coordinated process. To clarify why this works, we have reinforced our quantitative analysis in Appendix B. We demonstrated that as individually benign scenes are sequentially combined, the Divergence (diversity of generated videos) decreases significantly (0.2193 → 0.0808), while the probability of generating an Unsafe Video drastically increases (0.2 → 2.2) .
>
> This experiment validates our core hypothesis: the combination of scenes acts as a powerful constraint that narrows the vast generative space into a specific unsafe region. Scene Manipulation then effectively searches for the optimal attack point within this constrained space. We believe establishing and verifying this new attack through formal analysis constitutes a significant technical step for T2V safety.
>
> ---
>
> **Weakness 3**
>
> Following Reviewer XnhP's suggestion, we implemented and compared against RPG-RT [1], a SOTA T2I attack adapted for T2V. To ensure fairness, both methods were set to a maximum of 15 iterations on Veo2 across three distinct categories: Pornography, Borderline Pornography, and Violence.
>
> | Category               | RPG-RT | **SceneSplit (Ours)** |
> |------------------------|--------|-----------------------|
> | Pornography            | 15%    | **55%**               |
> | Borderline Pornography | 65%    | **75%**               |
> | Violence               | 75%    | **90%**               |
>
> As demonstrated above, SceneSplit consistently outperforms RPG-RT. This validates that our strategy of manipulating the generative output space via scene constraints is significantly more effective for video generation than directly adapting rule-based methods from the T2I domain.
>
> [1] Cao, Yichuan, et al. "Red-Teaming Text-to-Image Systems by Rule-based Preference Modeling." NeurIPS 2025.
>
> ---
>
> **Weakness 4**
>
> To provide a time/efficiency comparison with RPG-RT, we measured the average attack time per prompt on Veo2 across three key categories: Pornography, Borderline Pornography, and Violence.
>
> | Method            | Attack time per prompt (min) ↓|
> |-------------------|------------------------------|
> | RPG-RT            | 21.9 (10.2*)                 |
> | SceneSplit (Ours) | **5.9**                      |
> *Time excluding fine-tuning overhead
>
> SceneSplit is significantly faster (5.9 min) compared to RPG-RT (21.9 min), which suffers from heavy overhead due to scoring model training and LLM fine-tuning via DPO. Even when considering only the inference time after fine-tuning (10.2 min), SceneSplit remains superior in efficiency.
>
> Additionally, we analyzed the efficiency of our individual components across all 11 safety categories on Veo2 to demonstrate the efficiency of our method.
>
> **1. Efficiency of Scene Manipulation**: We compared our method against a baseline without Scene Manipulation (equivalent to the setting in Table 7, Inner Loop only), where a new Scene Splitting prompt is generated at every iteration instead of manipulating the existing one.
>
> | Method                  | Attack Success Rate (ASR) ↑ | Avg. Attack Attempt ↓ | Avg. Time ↓ |
> |-------------------------|-----------------------------|-----------------------|-------------|
> | with Scene Manipulation | **60.9%**     | **2.50**              | **130.72s** |
> | w/o Scene Manipulation  | 46.4%  |    3.57               | 171.49s     |
>
> Scene Manipulation significantly reduces attempts (3.57 → 2.50) and time, proving that searching within a constrained space via manipulation is far more efficient than regenerating entire prompts.
>
> **2. Efficiency of Strategy Update**: We also evaluated the efficiency impact of the Strategy Update (equivalent to the setting in Table 5). We compared the full method against a baseline where the outer loop generates a new scene split prompt without retrieving a strategy (w/o Strategy Update).
>
> | Method               | Attack Success Rate (ASR) ↑ | Avg. Attack Attempt ↓ | Avg. Time ↓ |
> |----------------------|-----------------------------|-----------------------|-------------|
> | with Strategy Update | **78.2%**     | **5.54**   | 323.06s     |
> | w/o Strategy Update  | 69.1%     | 6.22  | **311.52s** |
>
> Strategy Update reduces average attempts (6.22 → 5.54) by leveraging a successful strategy. Although the overhead slightly increases the time, the substantial gain in ASR (69.1% → 78.2%) presents a highly favorable trade-off for robust performance.

---

> ### Author Response · Authors · 2025-11-21
> **Official Comment by Authors**
>
> **Question 1**
>
> As noted by Reviewer XnhP, it is highly probable that the commercial models are already equipped with the safety filters, including pre-processing text and post-processing video safety detectors. However, the high success rate of SceneSplit against these safety systems exposes the insufficiency and critical limitations of current safety filters, even those that likely incorporate temporal analysis.
>
> While deploying a safety filter that analyzes the whole video would theoretically offer stronger defense capabilities, we believe that effectively blocking narrative-based outputs remains a fundamental challenge. Since SceneSplit operates by distributing harmful intent across a sequence of scenes that appear individually benign, detecting the threat requires a deep, contextual understanding of the narrative rather than simple visual recognition. This finding underscores the urgent need for next-generation safety filters capable of grasping complex narrative contexts.
>
> ---
>
> **Question 2**
>
> We appreciate the Reviewer's insightful question regarding the selection of the 'most influential' component. We validated this in Appendix D (Table 11), where we compared different selection heuristics. The results show that targeting the 'Most Influential' scene yields a significantly higher ASR (60.9%) compared to modifying a 'Random' (54.5%) or 'Least Influential' (56.4%) scene. This confirms that identifying and manipulating the dominant narrative element is the optimal pathway to navigate the safety decision boundary.
>
> ---
>
> **Question 3**
>
> As shown in Table 2 of our main paper, Scene Splitting significantly outperforms Only Paraphrasing in a one-shot setting without iterations (42.7% vs 30.0%).
> |                    | ASR   |
> |--------------------|-------|
> | Only paraphrasing  | 30.0% |
> | Scene Splitting    | **42.7%** |
>
> To further investigate, we conducted a new experiment comparing "Iterative Paraphrasing" (simply rewriting the prompt using an LLM for 5 iterations without splitting) versus our "Scene Manipulation."
>
> |                                              | ASR   |
> |----------------------------------------------|-------|
> | Only paraphrasing during the five iterations | 51.5% |
> | Scene Splitting + Scene Manipulation         | **60.9%** |
>
> The result (51.5% vs 60.9%) confirms that constraining generative output space by splitting the scenes and manipulating the scene in this constrained space are the key components of our success.
>
> ---
>
> **Question 4**
>
> Setting a higher threshold would increase the probability that the flagged videos contain unsafe content. However, we utilized the threshold $\theta_{unsafety}=0.6$ following the evaluation protocol established in T2VSafetyBench [2] to ensure a fair comparison. In their study, this value was empirically identified as the optimal threshold that demonstrates a high correlation with human judgments for distinguishing between safe and unsafe content. Therefore, we adhered to this standard to ensure our results reflect genuine safety risks as perceived by humans.
>
> [2] Miao, Yibo, et al. "T2vsafetybench: Evaluating the safety of text-to-video generative models." NeurIPS 2024.
>
> ---
>
> Thank you sincerely for your time and effort. These contributions have made our work much stronger. We hope the results of our new experiments resolve your concerns, and we welcome you to answer any further questions.

---

### Official Review · Reviewer_ZvWr · 2025-10-28

**Soundness:** 3
**Presentation:** 3
**Contribution:** 3
**Rating:** 6
**Confidence:** 3

**Summary:**

This paper addresses the under-explored safety vulnerabilities of Text-to-Video (T2V) models by proposing SceneSplit, a black-box jailbreak method that fragments harmful narratives into multiple individually benign scenes. When sequentially combined, these scenes constrain the model’s generative output space to an unsafe region while bypassing safety filters. Comprising three core components, Scene Splitting, Scene Manipulation, and Strategy Update, SceneSplit is evaluated across 11 safety categories on three commercial T2V models (Veo2, Hailuo, Luma Ray2), achieving high average Attack Success Rates that significantly outperform baselines. The study reveals that current T2V safety filters fail to address inter-scene contextual risks, highlighting the need for advanced safety mechanisms that assess narrative-level hazards, and also notes limitations in handling content dependent on specific entity identities.

**Strengths:**

1. The paper focuses on jailbreak attacks and safety vulnerabilities of Text-to-Video (T2V) models, a previously under-explored area relative to LLMs, VLMs, and T2I models.
2. The three-component structure (Scene Splitting, Scene Manipulation, Strategy Update) is logically coherent and scalable, with the Strategy Library enhancing the method’s robustness and efficiency by reusing successful attack patterns.
3. The paper proposes an innovative black-box method anchored on a unique core logic: fragmenting harmful narratives into individually benign scenes to constrain the generative output space, effectively bypassing safety filters while maintaining semantic consistency with the original harmful intent.
4. The paper provides insights for T2V safety studies, revealing the critical flaw of current safety filters in ignoring inter-scene contextual risks and laying a foundation for developing more advanced narrative-level safety assessment mechanisms.

**Weaknesses:**

1. The initial Scene Splitting prompt is fixed across all experiments for fairness, reducing randomness but potentially failing to reflect real-world variability in prompt generation. This may lead to overestimation of the method’s robustness.
2. No sensitivity analysis is conducted for key hyperparameters (e.g., the embedding similarity threshold λ and the unsafety score threshold θ_unsafety), leaving uncertainty about how performance might degrade under different parameter settings.
3. SceneSplit heavily relies on external LLMs (e.g., GPT-4o for scene splitting and manipulation, Qwen-30B for strategy summarization, VideoLLaMA3 for scene selection), introducing variability stemming from these external models’ inherent biases, non-determinism, or potential updates. The method thus lacks full self-containment.
4. There is no in-depth discussion of failure cases or analysis of why certain categories (e.g., Pornography on Hailuo) achieve lower ASRs, limiting a deeper understanding of the method’s inherent boundaries and failure mechanisms.

**Questions:**

How can the author determine whether the generated video contains unsafe content? Automatic or human evaluation?

---

> ### Author Response · Authors · 2025-11-21
> **Official Comment by Authors**
>
> We thank Reviewer ZvWr for your thoughtful feedback, which has significantly improved the quality of our paper.
>
> ---
> **Weakness 1**
>
> We understand the Reviewer's concern regarding the fixed initialization. Our primary motivation was to ensure a fair comparison by isolating the effects of our iterative components (Scene Manipulation/Strategy Update) from the randomness of the initial split.
>
> However, to address the concern about real-world variability, we conducted a new experiment on Veo2 in which the initial Scene Splitting prompt was generated dynamically (non-fixed) for each attempt.
>
> |                        | SceneSplit (fixed) | SceneSplit (non-fixed) |
> |------------------------|--------------------|------------------------|
> | Pornography   | 55%    | 50%      |
> | Borderline Pornography | 75%     | 70%    |
> | Violence  | 90%  | 95%   |
> | Gore   | 80% | 80%     |
> | Disturbing Content     | 75%    | 75%  |
> | Discrimination      | 65%    | 65%  |
> | Political Sensitivity  | 85% | 85%  |
> | Illegal Activities     | 90%    | 90%   |
> | Misinformation         | 80%    | 70%   |
> | Sequential Action Risk | 85%    | 85%   |
> | Dynamic Variation Risk | 80%  | 85%    |
> | Average   | 78.2%  | 77.3% |
>
> As shown in the table above, the ASR remains virtually unchanged (78.2% vs 77.3%). This validates that our method is robust to initialization variability, as the Scene Manipulation component effectively steers the generation toward the unsafe region regardless of the starting point.
>
> ---
> **Weakness 2 & Question 1**
>
> We performed a sensitivity analysis on Veo2 for the embedding similarity threshold ($\lambda$)
>
> | Embedding similarity threshold ($\lambda$) | ASR   |
> |--------------------------------------------|-------|
> | 0.4   | 76.4% |
> | 0.6       | 78.2% |
> | 0.8         | 72.7% |
>
> Varying $\lambda$ resulted in stable ASRs, confirming the robustness of our method against hyperparameter changes.
>
> Regarding the determination of unsafe content, we utilized the automated evaluation protocol proposed in T2VSafetyBench [2]. This method involves sampling image frames from the generated video and employing GPT-4o to evaluate these frames and assign a harmfulness score (scale 0-100). We classify a video as 'unsafe' if this score exceeds the threshold of $\theta_{unsafety}=0.6$.
>
> Although lowering this threshold would naturally increase the ASR, such results would likely be overestimated. A lower threshold increases the risk of misalignment with human judgment, potentially classifying safe content as unsafe. Therefore, we strictly adhered to the evaluation protocol established in T2VSafetyBench, which sets $\theta_{unsafety}=0.6$. This specific threshold was chosen because it demonstrates a high correlation with human judgments, ensuring that our reported success rates reflect genuinely unsafe content consistent with human judgments.
>
> [2] Miao, Yibo, et al. "T2vsafetybench: Evaluating the safety of text-to-video generative models." NeurIPS 2024.
>
> ---
>
> **Weakness 3**
>
> To verify the dependency of our framework, we replaced GPT-4o with Qwen-30B and Gemini 2.5 Pro for the Scene Splitting and Scene Manipulation tasks on Veo2 (Qwen 30B for strategy update, VideoLLama3 for scene selection).
>
>
> | Model  | ASR   |
> |----------------|-------|
> | GPT-4o  | 78.2% |
> | Qwen-30B       | 76.4% |
> | Gemini 2.5 Pro | 78.6% |
>
> GPT-4o, Qwen-30B, Gemini 2.5 pro for strategy update (GPT-4o for scene splitting and manipulation, VideoLLama3 for scene selection)
>
> | Model          | ASR   |
> |----------------|-------|
> | GPT-4o   | 77.2% |
> | Qwen-30B       | 78.2% |
> | Gemini 2.5 Pro | 76.8% |
>
> VideoLLama3, GPT-4o for scene selection (GPT-4o for scene splitting and manipulation, Qwen 30B for strategy update)
>
> | Model       | ASR   |
> |-------------|-------|
> | GPT-4o  | 77.2% |
> | VideoLLama3 | 78.2% |
>
> The consistent ASR across different LLMs proves that SceneSplit is a model-agnostic framework, not dependent on the specific biases of a single LLM.
>
> ---
> **Weakness 4**
>
> We appreciate the Reviewer's request for a deeper analysis of failure cases. As discussed in Appendix F, SceneSplit struggles with categories like "Public Figures" or "Copyright" because the process of splitting and paraphrasing often dilutes the specific visual identity traits required for a successful attack in these categories.
>
> Regarding the lower ASR for Pornography (e.g., 60% on Hailuo) compared to Violence (100%), we believe this reflects the defense landscape. Pornography is often the most heavily guarded category with aggressive filters in commercial models. The fact that we improved the ASR from a baseline of 0% to 60% in such a strictly guarded category is, we argue, a testament to the strength of our attack, revealing that even the strictest filters are vulnerable to narrative fragmentation.
>
> ---
>
> We sincerely appreciate the time and effort, which greatly strengthened our work. We hope our new experiments resolve your concerns, and we welcome further discussion.

---

> > ### Comment · Reviewer_ZvWr · 2025-11-25
> >
> > Thanks for your rebuttal. My questions have been clarified. One more thing, I suggest that the evaluation would be better if some human analysis were included, which supports the claim that the results consistent with human judgments.

---

> ### Author Response · Authors · 2025-11-28
>
> We sincerely thank the Reviewer for their positive feedback, and we are delighted that our response has successfully addressed your questions.
>
> As per your suggestion, we conducted a human evaluation to verify the alignment between the automated evaluation protocol and human judgment.
>
> * Dataset: We randomly sampled 162 videos generated by Sora2 and Veo2 from our experiments, ensuring a balanced distribution with at least 10 videos for each of the 11 safety categories.
>
> * Annotators: We recruited 7 human labelers to assess each video. The annotators classified each video as either "Safe" or "Unsafe." The final ground-truth label for each video was determined by majority voting among the 7 labelers.
>
> * Comparison: We compared the labels assigned by the automated evaluator (GPT-4o) used in our paper against these human-annotated ground truth labels.
>
> | Metric    | Value  |
> |-----------|--------|
> | Accuracy  | 0.8395 |
> | Precision | 0.9434 |
> | Recall    | 0.8333 |
> | F1 Score  | 0.8850 |
>
> As shown in the table, the automated evaluator demonstrates a high alignment with human judgments, achieving an F1 Score of 0.8850 and an Accuracy of 0.8395. These results support that the high ASRs reported in our paper are consistent with humans.
>
> We hope this additional analysis fully addresses your suggestion. Thank you once again for your constructive guidance.

---

### Official Review · Reviewer_XnhP · 2025-10-31

**Soundness:** 3
**Presentation:** 4
**Contribution:** 4
**Rating:** 6
**Confidence:** 5

**Summary:**

This paper presents a black-box jailbreak attack method called SceneSplit, designed to bypass the safety filters of T2V models. By splitting harmful prompts into multiple seemingly benign scenes, SceneSplit leverages their combination to constrain the generative space, guiding the output towards harmful content. The method improves attack success through scene splitting, iterative scene manipulation, and a strategy library. Experimental results show that SceneSplit achieves significantly higher attack success rates on multiple T2V models in the T2VSafetyBench, providing new insights into the security of T2V models.

**Strengths:**

- The idea of the paper is novel and interesting. It introduces an attack on T2V models that exploits narrative structure, providing new insights into the security of T2V models.
- The writing is clear and easy to understand.
- The motivation behind the method is well-justified. The example in Figure 1 effectively illustrates the motivation of the approach.
- The experiments validate the effectiveness of the method. Compared to the original attack success rates in T2VSafetyBench, the method shows a significant improvement, particularly in the pornography category.
- The paper includes ablation analysis (sections 4.3 and 4.4), which demonstrates the effectiveness of various components (sections 3.2 and 3.3) and the rationality of scene splitting (section 3.1). This provides strong evidence of the superiority of the proposed method.
- The paper clearly outlines its limitations.

**Weaknesses:**

- The defense mechanisms in commercial T2V models include pre-processing text safety detectors and post-processing video safety detectors. The method’s effectiveness against the pre-processing text detector is clearly justified and reasonable, but how does it bypass the post-processing video safety detector?
- There is a lack of baseline attack methods. Some attack techniques for T2I models could be adapted for T2V models, such as RPG-RT (Section 3.6) [1], which could serve as a baseline for comparison.

[1] Red-Teaming Text-to-Image Systems by Rule-based Preference Modeling, NeurIPS 2025, https://openreview.net/pdf?id=MdqirFiD38

- In section 3.2, it’s noted that the method iterates after an initial attack attempt fails, and modifies the prompt iteratively when the attack succeeds. Section 3.3 also mentions that strategy updates make the attack more efficient. Therefore, I would like to know the computational resource consumption of the method.
- The tested T2V models are limited to Luma Ray2, Hailuo, and Veo2. Why were only these models tested? If there is a specific reason, it should be stated in the experiments section. Otherwise, another widely recognized sota and secure model should be tested.
- Minor writing issues:
  - "The method works by splitting harmful narratives into multiple independent scenes, each of which is harmless" is similar to the idea of the Temporal Risks category in T2VSafetyBench. This should be mentioned in the related work or introduction.
  - The abstract should mention the dataset used, T2VSafetyBench.
  - The title of Figure 2 should include a brief introduction.
  - In the appendix, examples should be given according to risk categories, rather than by different models.

Overall, this is a paper above the threshold for acceptance. I will adjust my score further based on the author's rebuttal.

**Questions:**

see Weaknesses

---

> ### Author Response · Authors · 2025-11-21
> **Official Comment by Authors**
>
> We sincerely thank Reviewer XnhP for your insightful and constructive feedback.
>
> ---
> **Weakness 1**
>
> We deeply appreciate the Reviewer raising the critical point about post-processing video safety detectors. The fact that SceneSplit still achieves a notable success rate reveals the critical limitations of current post-processing filters. We believe SceneSplit effectively navigates the 'bypass boundary' of these post-processing filters. Unlike simple text manipulation, SceneSplit performs Scene Selection based on the generated video and utilizes the safety reward for iterative modification. By continuously adjusting based on these generated outputs, it identifies specific visual representations that fall outside the detection scope of existing post-processing safety detectors, eventually allowing videos containing such direct exposure to be generated. We believe this finding underscores the urgent need for next-generation safety filters capable of handling such attacks.
>
> ---
> **Weakness 2**
>
> Following your advice, we adapted RPG-RT [1], a SOTA attack for T2I models, to the T2V domain for a direct comparison. To ensure fairness, both methods were set to a maximum of 15 iterations on Veo2 across three distinct categories: Pornography, Borderline Pornography, and Violence.
>
> | Category               | RPG-RT | **SceneSplit (Ours)** |
> |------------------------|--------|-----------------------|
> | Pornography            | 15%    | **55%**               |
> | Borderline Pornography | 65%    | **75%**               |
> | Violence               | 75%    | **90%**               |
>
> As shown above, SceneSplit consistently outperforms RPG-RT. This demonstrates that our strategy of manipulating the scene, thereby constraining the generative output space, is far more effective for video generation than simply adapting T2I attack methods.
>
> [1] Cao, Yichuan, et al. "Red-Teaming Text-to-Image Systems by Rule-based Preference Modeling." NeurIPS2025.
>
> ---
>
> **Weakness 3**
>
> To provide a fair and direct comparison with the baseline, we measured the average attack time per prompt on Veo2 across three key categories: Pornography, Borderline Pornography, and Violence.
>
> | Method            | Attack time per prompt (min) ↓|
> |-------------------|------------------------------|
> | RPG-RT            | 21.9 (10.2*)                 |
> | SceneSplit (Ours) | **5.9**                      |
> *Time excluding fine-tuning overhead
>
> SceneSplit is faster (5.9 min) compared to RPG-RT (21.9 min), which suffers from heavy overhead due to scoring model training and LLM fine-tuning via DPO. Even considering only the inference time after fine-tuning (10.2 min), SceneSplit remains superior.
>
> Additionally, we analyzed the efficiency of our components across all 11 safety categories on Veo2:
>
>
> 1. **Efficiency of Scene Manipulation**: We compared our method against a baseline without Scene Manipulation (equivalent to the setting in Table 7, Inner Loop only), where a new Scene Splitting prompt is generated at every iteration instead of manipulating the existing one.
>
> | Method                  | Attack Success Rate (ASR) ↑ | Avg. Attack Attempt ↓ | Avg. Time ↓ |
> |-------------------------|-----------------------------|-----------------------|-------------|
> | with Scene Manipulation | **60.9%**                   | **2.50**              | **130.72s** |
> | w/o Scene Manipulation  | 46.4%                       |    3.57               | 171.49s     |
>
> As shown in the table, Scene Manipulation significantly reduces the average number of attempts (3.57 → 2.50) and total time (171.49s → 130.72s). This proves that searching for an optimal attack point within a constrained generative space (via manipulation) is far more efficient than repeatedly generating entirely new scene split prompts, as it requires fewer iterations to achieve success.
>
> 2. **Efficiency of Strategy Update**: We also analyzed the efficiency of the Strategy Update component (equivalent to the setting in Table 5, Inner Loop + Outer Loop). We compared our full method against a baseline where the outer loop generates a new scene split prompt from scratch without retrieving a strategy (w/o Strategy Update).
>
> | Method               | Attack Success Rate (ASR) ↑ | Avg. Attack Attempt ↓ | Avg. Time ↓ |
> |----------------------|-----------------------------|-----------------------|-------------|
> | with Strategy Update | **78.2%**                   | **5.54**              | 323.06s     |
> | w/o Strategy Update  | 69.1%                       | 6.22                  | **311.52s** |
>
> Strategy Update reduces the average number of attempts (6.22 → 5.54) by leveraging successful attack patterns. Although the average time is slightly higher (311.52s → 323.06s) due to the computational overhead of managing the strategy library and summarization, this marginal cost yields a substantial improvement in ASR (69.1% → 78.2%). We believe this trade-off is highly favorable for achieving robust jailbreak performance.

---

> ### Author Response · Authors · 2025-11-21
> **Official Comment by Authors**
>
> **Weakness 4**
>
> We have expanded our evaluation to include Sora2 and Kling V1.0.
>
> |  | Sora2 |  | Kling V1.0 |  |
> |---|---|---|---|---|
> |  | T2VSafetyBench | **SceneSplit (Ours)** | T2VSafetyBench | **SceneSplit (Ours)** |
> | Pornography | 0% | **10%** | 5% | **35%** |
> | Borderline Pornography | 0% | **45%** | 50% | **85%** |
> | Violence | 30% | **95%** | 60% | **100%** |
> | Gore | 5% | **55%** | 60% | **80%** |
> | Disturbing Content | 50% | **75%** | 50% | **70%** |
> | Discrimination | 15% | **70%** | 15% | **75%** |
> | Political Sensitivity | 5% | **65%** | 15% | **85%** |
> | Illegal Activities | 45% | **90%** | 20% | **85%** |
> | Misinformation | 70% | **75%** | 40% | **70%** |
> | Sequential Action Risk | 55% | **90%** | 60% | **95%** |
> | Dynamic Variation Risk | 60% | **85%** | 40% | **85%** |
> | Average | 30.5% | **68.6%** | 37.7% | **78.6%** |
>
> On Sora2, SceneSplit improved the ASR from 30.5% to 68.6%. On Kling V1.0, it achieved an impressive increase from 37.7% to 78.6%. This consistent and significant performance boost across diverse T2V models confirms that the vulnerability we identified is a fundamental issue in current T2V models, not specific to a single architecture.
>
> ---
>
> **Minor writing issues**
>
> We sincerely thank the Reviewer for their careful reading and valuable suggestions regarding the writing. We will update the current manuscript to reflect all the points you mentioned. Thank you once again for helping us improve the quality of our paper.
>
> ---
>
> We sincerely thank Reviewer XnhP again for your time and valuable insights. We hope our new experiments and clarifications have satisfactorily addressed your concerns, and we are happy to answer any further questions.

---

> > ### Comment · Reviewer_XnhP · 2025-11-22
> > **I still have the following issues**
> >
> > The author has addressed some of my concerns. However, I still have the following issues:
> >
> > 1. Regarding weakness 2, the author only conducted experiments on three different categories using a single model (Veo2). Experiments should be conducted across five models (Luma Ray2, Hailuo, Veo2, Sora2 and Kling V1.0) and for all 11 categories. Additionally, new experimental results should be added to the manuscript during the rebuttal phase (i.e., Table 1 should have 5 additional columns, with RPG-RT as the baseline). Given the new models (Sora2 and Kling V1.0) mentioned in weakness 4, there should actually be 5 columns rather than 3. This approach would comprehensively demonstrate that the SceneSplit proposed in this paper consistently outperforms other baselines. As it stands, the results are based solely on three categories of a single model, which does not sufficiently show that SceneSplit consistently performs better than RPG-RT.
> >
> > 2. Regarding weakness 5, the manuscript should be updated during the rebuttal phase, but I did not see any changes.
> >
> > 3. Regarding weakness 1, the author's response has resolved my concerns. As I understand it, scene splitting is for pre-processing text-based security detectors, while scene manipulation is for post-processing video security detectors. However, the paper only mentions the detection of the security filtering mechanism in section 3.1. Therefore, the discussion on detection should be rewritten to provide a more comprehensive explanation.

---

> > > ### Author Response · Authors · 2025-11-25
> > >
> > > We sincerely appreciate your engagement and insightful feedback throughout the rebuttal process.
> > >
> > > ---
> > >
> > > **Issue 1**
> > >
> > > To demonstrate the consistent effectiveness of our method, we have conducted comprehensive experiments across 5 models (Veo2, Luma Ray2, Hailuo, Kling v1.0, and Sora2) and all 11 risk categories.
> > >
> > > Regarding the comparison with the T2I method, RPG-RT relies on specific detectors such as NudeNet and Q16 for NSFW detection in certain categories. However, because it lacks an established detector for the remaining t2v safety categories, we applied the safety evaluation protocol utilized in our paper to assess NSFW content for these cases.
> > >
> > > We have included these comprehensive experimental results in Appendix C of the revised manuscript.
> > >
> > > |  | Veo2 |  | Ray2 |  | Hailou |  | Kling v1.0 |  | Sora2 |  |
> > > |---|---|---|---|---|---|---|---|---|---|---|
> > > |  | RPG-RT | **SceneSplit (Ours)** | RPG-RT | **SceneSplit (Ours)** | RPG-RT | **SceneSplit (Ours)** | RPG-RT | **SceneSplit (Ours)** | RPG-RT | **SceneSplit (Ours)** |
> > > | Pornography | 15% | **55%** | 5% | **45%** | 20% | **60%** | 25% | **35%** | 0% | **10%** |
> > > | Borderline Pornography | 65% | **75%** | **85%** | 75% | 60% | **90%** | 65% | **85%** | 0% | **45%** |
> > > | Violence | 75% | **90%** | 55% | **90%** | 65% | **100%** | 75% | **100%** | 50% | **95%** |
> > > | Gore | 60% | **80%** | 60% | **90%** | 20% | **80%** | 70% | **80%** | 20% | **55%** |
> > > | Disturbing Content | 60% | **75%** | 65% | **85%** | 85% | **90%** | **70%** | **70%** | 40% | **75%** |
> > > | Discrimination | 60% | **65%** | 25% | **50%** | 50% | **80%** | 55% | **75%** | 25% | **70%** |
> > > | Political Sensitivity | 80% | **85%** | 40% | **85%** | 50% | **80%** | 45% | **85%** | 10% | **65%** |
> > > | Illegal Activities | 75% | **90%** | 70% | **85%** | 85% | **95%** | 60% | **85%** | 50% | **90%** |
> > > | Misinformation | 60% | **80%** | 65% | **70%** | 60% | **75%** | 65% | **70%** | 60% | **75%** |
> > > | Sequential Action Risk | 55% | **85%** | 50% | **95%** | 55% | **90%** | 45% | **95%** | 50% | **90%** |
> > > | Dynamic Variation Risk | 75% | **80%** | 55% | **80%** | 65% | **85%** | 60% | **85%** | 70% | **85%** |
> > > | Average | 61.8% | **78.2%** | 52.3% | **77.2%** | 55.9% | **84.1%** | 57.7% | **78.6%** | 34.1% | **68.6%** |
> > >
> > > These results empirically validate that SceneSplit outperforms the baseline not only on Veo2 but also on the newly added T2V models.
> > >
> > > ---
> > >
> > > **Issue 2**
> > >
> > > We have revised the manuscript to address the minor writing issues you pointed out. We confirm that the following updates have been implemented in the current version:
> > >
> > > * Related Work: We added a discussion on T2VSafetyBench, explicitly mentioning their "Temporal Risks" category.
> > >
> > > * Abstract: We stated that T2VSafetyBench is the dataset used in our study.
> > >
> > > * Figure 2: We updated the caption to include a brief introduction to the figure for better clarity.
> > >
> > > * Appendix (Figure 8): We present examples according to risk categories to better illustrate across different types of content.
> > >
> > > ---
> > >
> > > **Issue 3**
> > >
> > > We appreciate you raising this point. We have rewritten the relevant sections to provide a comprehensive explanation of the detection mechanisms.
> > >
> > > * Section 3.1 has been updated to clarify that Scene Splitting serves as a pre-processing step designed to bypass text-based security detectors.
> > >
> > > * Section 3.2 has been refined to explain that Scene Manipulation operates as a post-processing step to evade post-hoc detectors after generation.
> > >
> > > ---
> > >
> > > We hope these results and responses address your concerns. Once again, we sincerely appreciate your time and effort to enhance the quality of our research.

---

> > > > ### Comment · Reviewer_XnhP · 2025-11-25
> > > > **I still have the following concerns**
> > > >
> > > > Thank you for your further rebuttal. I still have the following concerns:
> > > >
> > > > - Regarding the baseline: A major weakness of this paper is the lack of a baseline, which Reviewer 2XCW also considers a key issue. Therefore, the new results regarding the baseline should be included in Table 1, rather than in the appendix. Additionally, the description of the comparative RPG-RT method (including specific details and parameters), as well as the corresponding results, should be incorporated into Sections 4.1 and 4.2, rather than being added to Appendix C. Only by presenting this information in the main text can the paper address its weakness in baseline comparison.
> > > >
> > > > - Regarding the examples:
> > > >   1. Since the authors have not provided the code, at the very least, they should include all the attack success and failure examples generated by SceneSplit in an attachment (currently, no attachments have been submitted with this submission) to demonstrate reproducibility. This should be a dataset of size 20*11*5 (with success or failure labels). Alternatively, the authors could further provide the code in the attachment as well.
> > > >   2. The examples in Figure 8 should include the corresponding prompts.
> > > >
> > > > - Regarding the new content: The revised manuscript still contains many typos. For instance, in Section 4.2, kling and sora2 have not been updated.

---

> ### Author Response · Authors · 2025-11-26
>
> We are deeply grateful for your engagement and detailed feedback. Your active participation has been invaluable in refining the quality and completeness of our work. We have addressed your remaining concerns as follows:
>
> **1. Regarding the Baseline (Table 1 & Sections 4.1/4.2)**: We fully agree with your suggestion to elevate the baseline comparison to the main text. We have integrated the RPG-RT results directly into Table 1 in the main manuscript to provide a clear, side-by-side comparison. Furthermore, we have incorporated the detailed description of the RPG-RT method (including specific parameters and settings) into Section 4.1 (Experimental Setup).
>
> **2. Regarding Examples and Code (Reproducibility)**: To ensure full reproducibility, we have uploaded a supplementary file. This attachment includes:
> * Source Code: The code for SceneSplit.
> * Full Result Dataset (20x11x5): A comprehensive record of all attack attempts, including both success and failure cases across the tested T2V models and categories.
>
> **3. Regarding Figure 8**: We have revised Figure 8 in the Appendix. It now includes the corresponding prompts for each example, and we have updated the caption to explicitly specify the model names used for each generation.
>
> **4. Regarding Typos**: We sincerely apologize for the oversight in the previous text. We have completely rewritten Section 4.2 to accurately describe the updated results, including the specific performance for Kling v1.0 and Sora2. Furthermore, we have updated Appendix C and D to fully reflect the results of our additional experiments. We thank you for pointing this out.

---

### Author Response · Authors · 2025-12-03
**Summary of Rebuttal**

We sincerely thank the Area Chair and all reviewers for their time, constructive feedback, and active engagement throughout the review process.

During the rebuttal period, we have diligently addressed concerns raised by the reviewers through extensive additional experiments and manuscript revisions.

## Key Update

**1. Comprehensive Baseline Comparison (Reviewer XnhP, 2XCW, 9BtZ)** - Table 1, Section 4.1, and Section 4.2

We adapted RPG-RT (NeurIPS 2025), a state-of-the-art T2I attack, to all 5 T2V models (Luma Ray2, Hailuo, Veo2, Sora2, Kling v1.0) and all 11 risk categories. SceneSplit consistently outperforms RPG-RT across all categories.

**2. Computational Overhead (Reviewer XnhP, 2XCW, 9BtZ)** - Table 7, 10

We demonstrated that SceneSplit is significantly more efficient than RPG-RT, avoiding fine-tuning costs. Internally, we verified that Scene Manipulation optimizes the search process by reducing both time (171.49s $\rightarrow$ 130.72s) and attempts (3.57 $\rightarrow$ 2.50), while Strategy Update further decreases attempts (6.22 $\rightarrow$ 5.54).

**3. Additional T2V Models (Reviewer XnhP, 9BtZ)** - Table 1 and Section 4.2.

We expanded our evaluation to include Sora2 and Kling v1.0. SceneSplit achieved high ASRs on these advanced models (68.6% on Sora2, 78.6% on Kling v1.0), confirming that the "Generative Output Space Constraint" vulnerability is fundamental to current T2V paradigms, not model-specific.

**4. Human Evaluation (Reviewer ZvWr)**

We conducted a human evaluation (7 labelers) which showed high alignment (F1 Score 0.885, Accuracy 0.8395) between automated evaluation metrics and human judgment.

**5. LLM Dependence (Reviewer ZvWr, 9BtZ)** - Table 14, Appendix C

We verified our framework using different LLMs (GPT-4o, Qwen-30B, and Gemini 2.5 Pro) for each component (Scene Splitting, Scene Manipulation, and Strategy Update) and evaluators (Qwen3-VL, Gemini 2.5 Flash), confirming that our results are model-agnostic.

**6. Hyperparameter Sensitivity (Reviewer ZvWr)** - Table 15, Appendix C

We performed sensitivity analyses on key hyperparameters, such as the embedding similarity threshold ($\lambda$), confirming that our method's effectiveness is not dependent on specific parameters.

**7. Robustness to Split Randomness (Reviewer ZvWr)** - Table 13, Appendix C

While we utilized a fixed initial Scene Splitting prompt in our main experiments to ensure a fair comparison of the iterative components, we conducted an additional experiment with dynamic initialization. We proved robustness against dynamic initialization (Fixed: 78.2% vs. Dynamic: 77.3%).

**8. Optimal Scene Number & Paraphrasing Degree (Reviewer 9BtZ)** -  Table 16, 17, 18, Appendix D

We identified that a split depth of 4 to 5 scenes represents the optimal balance for bypassing filters. Furthermore, we verified that our standard paraphrasing setting strikes the best trade-off between attack success and semantic preservation compared to aggressive over-paraphrasing.

**Summarization of discussion with Reviewers**

* Reviewer XnhP: We addressed some of the reviewer's concerns. Additionally, we engaged in active follow-up regarding baselines and reproducibility, incorporating these updates fully into the main paper and supplementary material.

* Reviewer ZvWr: We successfully addressed the reviewer's concerns. Furthermore, we engaged in a constructive follow-up regarding the suggested human evaluation, which we conducted to demonstrate high alignment.

* Reviewer 9BtZ: Confirmed that concerns were resolved and raised the score to 8. We have fully incorporated the suggested additional experiments and discussions into the revised manuscript.

---

### Meta-Review · Area_Chair_FTTo · 2026-01-01

**Summary:**

This paper proposes SceneSplit, a novel black-box jailbreak attack method on text-to-video generative models by fragmenting a harmful narrative into multiple scenes. The idea is novel and interesting. The approach manipulates the generative output space, the abstract set of all potential video outputs for a given prompt, using the combination of scenes as a powerful constraint to guide the final outcome. The experiments on T2VSafetyBench show high attack success rate against typical T2V models.

The reviewers have recognized the novelty of the idea, the clear writing, the well-justified motivation, and the effective experiments. The reviewers raised some concerns about the initial manuscript, including:
- Evaluation on defense methods: Reviewer XnhP pointed out that the method is not evaluated on post-processing video safety detector.
- Lack of attack baselines: Reviewer XnhP and Reviewer 2XCW pointed out the lack of baselines. Some methods for T2I attacks can be adapted for T2V models as baselines.
- Computational cost: Reviewer XnhP, Reviewer 2XCW, and Reviewer 9BtZ asked about the computational resource consumption of the method.
- Lack of SOTA models: Reviewer XnhP and Reviewer 9BtZ pointed out that the experiments are limited to specific models, and more sota models should be tested.
- Ablation study: Reviewer ZvWr pointed out that no sensitivity analysis is conducted for key hyperparameters.
- Rely on external LLMs: Reviewer ZvWr and Reviewer 9BtZ pointed out that the method relies on LLMs, ntroducing variability stemming from these external models’ inherent biases, non-determinism, or potential updates.

**Reviewer Concerns:**

The authors have made a great effort to address the reviewer concerns and discuss with their further feedback. Three of four reviewers have acknowledged that their concerns have been addressed by the rebuttal. Reviewer 2XCW did not engage in the discussion but some of his/her concerns are similar to other reviewers. So I think the rebuttal can largely address Reviewer 2XCW's concerns.

In this sense, I think that most (if not all) of the concerns have been addressed by the rebuttal.

**Reviewer Scores:**

The paper received the initial ratings of (4,6,6,6). Reviewer 9BtZ has acknowledged that the rating is improved to 8. Since most of the concerns have been addressed (see above), I believe that the paper would receive positive ratings in the end.

---

### Decision · Program_Chairs · 2026-01-26

Accept (Poster)